# OpenReview forum: "LinAJD: Scalable Gradient-Free Jailbreak Defense via Linearly Separable Embeddings"
_ICLR.cc/2026/Conference — ICLR 2026 Conference Withdrawn Submission_

### Official Review · Reviewer_Cg8Z · 2025-10-31

**Soundness:** 3
**Presentation:** 3
**Contribution:** 2
**Rating:** 2
**Confidence:** 3

**Summary:**

In this paper, the authors propose LinAJD, the Linear Adversarial Jailbreak Defense for large language models. In contrast to previous works, they craft perturbations based on the linearity of the safety bound. Extensive experiments on multiple open-source models indicate the efficiency and effectiveness of the proposed method.

**Strengths:**

1 The soundness of the method is good.

2 The writing is easy to follow.

3 The experiments are relatively solid.

**Weaknesses:**

1 The categorization of related work needs to be optimized. In Section 2, the authors attribute the SmoothLLM defense as one variant of adversarial training for the jailbreak defense. However, as stated in this paper, SmoothLLM is a defense that “perturbs multiple versions of a given prompt and aggregates their outputs to detect adversarial behavior through consensus.” With this definition, it seems that it is an input preprocessing defense instead of adversarial training.

2 With the proceeding of the training process, the embedding features of the target models could diverge from the original distribution that the pretrained classifiers are trained on. This could explain why in adversarial training, it usually requires generating adversarial samples on-the-fly instead of prior to the training. Thus, adopting the pretrained classifier could weaken the defense.

3 The evaluations are performed against the optimization-based attacks, such as GCG and SCAV which adversarial training is good at dealing with. However, in addition to optimization-based attacks, query-based attacks [1-3] and context-based attacks [3-4] are demonstrated to be powerful in reality. Authors should perform experiments across these attacks to demonstrate the generalization of the proposed method.

4 As claimed in this paper, the LinAJD achieves a good trade-off between efficiency and performance. Contrastly, PAT in [5] and RPO in [6] propose to perform adversarial training on the input prompts, making it more efficiently and transferable to the closed-source model. I recommend that authors discuss both of them in the related work section and select them as baselines.


[1] Jailbreaking black box large language models in twenty queries

[2] Tree of attacks: Jailbreaking black-box llms automatically

[3] AutoDAN-Turbo: A Lifelong Agent For Strategy Self-exploration to Jailbreak LLMs

[3] Jailbreak and guard aligned language models with only few in-context demonstrations

[4] Improved few-shot jailbreaking can circumvent aligned language models and their defenses

[5] Fight Back Against Jailbreaking via Prompt Adversarial Tuning

[6] Robust prompt optimization for defending language models against jailbreaking attacks

**Questions:**

1 Noticing that experiments are performed on LLMs of smaller scales, such as 7B and 8B, will it work for models of larger scales, such as Llama2-13B or Qwen2.5-32B?

2 Why adopt the DPO and SimPO as the objectives for learning? Why not directly fine-tune the target models with supervised fine-tuning (SFT)?

---

> ### Author Response · Authors · 2025-11-21
> **Responses to the main weaknesses**
>
> Thank you very much for taking the time to review our paper and for your comments on the method, writing, and experiments. We appreciate your feedback, and we address your concerns and points of disagreement in detail below.
>
> **Weakness 1 – Categorization of SmoothLLM in related work**
>
> Thank you very much for pointing this out. You are right that SmoothLLM, as originally defined, perturbs multiple versions of a given prompt at inference time and aggregates their outputs via consensus, which is closer to an inference-time input/representation smoothing defense than to training-time adversarial training. In the revision, we will adjust our taxonomy accordingly, explicitly categorize SmoothLLM as an inference-time defense rather than an adversarial-training variant, and clarify the distinction between training-time and inference-time defenses in Section 2.
>
> **Weakness 2 – Potential mismatch between pretrained classifiers and evolving embeddings**
>
> Thank you for raising this concern. We have explicitly investigated this issue in **Section 4.3.4**, where we compare LinAJD with **fixed** classifiers against a variant that **updates the classifiers on-the-fly** during training. As you can see there, updating the classifiers jointly with the model in fact degrades robustness. Our interpretation is that this degradation is caused by unstable classifier boundaries during joint optimization: as the model’s representations evolve, the inferred adversarial direction and magnitude become unreliable, which undermines the effectiveness of the defense. We will make this connection to your concern more explicit in the revised text and point readers directly to these results.
>
> **Weakness 3 – Evaluation beyond optimization-based attacks**
>
> Thank you for this suggestion. Due to time and computation constraints, we focused our additional experiments on a subset of the query/context–style attacks you mention. On **Llama2-7B**, we added results for **PAIR** and **I-FSJ**, both of which are semantically rich, query-based/context-driven jailbreaks; as shown in **Table R1**, LinAJD-S still substantially reduces ASR and A-Score.
>
> **Table R1: Results on additional  attacks (Llama2-7B)**
>
> | Attack | Llama2-7B / ASR | Llama2-7B / A-Score | LinAJD-S / ASR | LinAJD-S / A-Score |
> |--------|---------------|-------------------|---------------------|-------------------------|
> | I-FSJ  | 98%           | 88.95%            | 58%                 | 52.35%                  |
> | PAIR   | 66%           | 79.04%            | 0%                  | 0%                     |
>
> To also address your concern about larger models, we report **Qwen2.5-14B** results in **Table R2**, where LinAJD-S consistently improves robustness against PAIR, LAA, DRA, and SCAV while maintaining reasonable A-Scores. We did not run I-FSJ on Qwen2.5-14B because, in our experiments, it did not reliably succeed even on the base model.
>
> **Table R2: Results on attacks for Qwen2.5-14B and LinAJD-S**
> | Attack | Qwen2.5-14b/ASR | Qwen2.5-14b/A-Score | LinAJD-S/ASR |  LinAJD-S/A-Score |
> |--------|-----------------|---------------------|----------------|--------------------|
> | PAIR   | 90%             | 87.78%              | 38%            | 67.98%             |
> | LAA    | 100%            | 53.50%              | 4%             | 33.30%             |
> | DRA    | 86%             | 68.80%              | 0%             | 0%                 |
> | SCAV   | 84%             | 84.92%              | 6%             | 63.89%             |
>
> **Weakness 4 – Comparison with PAT and RPO**
>
> Thank you for this suggestion. We will add a discussion of PAT [5] and RPO [6] in the related work. However, their setting is fundamentally different from ours: LinAJD fine-tunes the **model itself** to be robust against arbitrary future attackers, whereas PAT and RPO build **prompt/suffix-level wrappers** around a (often closed-source) model. If an attacker can directly query the underlying model or strip the protective suffix, PAT/RPO can be bypassed, while LinAJD is designed precisely for this scenario. We will clarify this difference in threat models and position PAT/RPO as complementary rather than directly comparable baselines.
>
> **References**
>
> PAIR: Chao et al. Jailbreaking Black Box Large Language Models in Twenty Queries. arXiv 2023.
>
> I-FSJ: Zheng X et al. Improved few-shot jailbreaking can circumvent aligned language models and their defenses. NeurIPS 2024.

---

> ### Author Response · Authors · 2025-11-21
> **Responses to the questions**
>
> Thank you for raising these questions; we are happy to discuss them in more detail and address each one below.
>
> **Question 1 – Larger-scale models**
>
> Thank you for this question. We have addressed it  in our response to **Weakness 3** above and kindly refer you there for details and the new large-model results.
>
> **Question 2 – Why DPO/SimPO instead of SFT**
>
> Thank you for raising this. LinAJD is conceptually decoupled from the specific training objective: it provides a linear preference/perturbation signal in embedding space that could, in principle, be combined with SFT, DPO, SimPO, or other objectives. In this paper, we instantiate it with **DPO-style** (LinAJD-D) and **SimPO-style** (LinAJD-S) losses mainly because (i) they align with the objectives used by our baselines, allowing us to reuse their hyperparameters and ensure fair, comparable training, and (ii) these objectives are already known to be effective for safety alignment. Adding an additional training variant (for example, SFT+LinAJD) is quite expensive in our setup, since some attacks become very costly against defended models (for instance, **LAA** takes about **558 s** on the base model, **7056 s** on CAPO, and **15132 s** on a LinAJD-tuned model). Given these constraints, we focused on DPO/SimPO for this submission, but we agree that combining LinAJD with SFT and other objectives is interesting and we plan to explore this in future work.

---

> > ### Comment · Reviewer_Cg8Z · 2025-11-27
> >
> > Dear authors,
> >
> > We appreciate your rebuttal. As there is still time to incorporate the reviewers' feedback, we encourage you to submit a revised version of this manuscript.
> >
> > Thank you so much!
> >
> > Reviewer Cg8Z

---

> > > ### Author Response · Authors · 2025-11-28
> > >
> > > Thank you again for your thoughtful comments and for pointing out these important related directions. In the revised version, we have (i) expanded the Related Work section with a new paragraph on **prompt-level defenses**, where we explicitly discuss Prompt Adversarial Tuning, Robust Prompt Optimization, and SmoothLLM and clarify how our approach differs from and complements these methods; (ii) added additional analyses in **Section 4.2** to better explain the supplementary experiments, with the full experimental results moved to **Appendix C**; and (iii) included the missing training details for the Qwen2.5-14B model in **Table 8 in Appendix D.3**. We hope these revisions address your concerns, and we are grateful for your suggestions that helped us improve the clarity and completeness of the paper.

---

> > > > ### Comment · Reviewer_Cg8Z · 2025-11-28
> > > >
> > > > Dear authors,
> > > >
> > > > Thank you so much for your rebuttal. My concerns are well addressed. However, I am currently unable to modify the scores. Once the OpenReview system is restored, I will substantially increase my ratings.
> > > >
> > > > Best wishes,
> > > >
> > > > Reviewer Cg8Z

---

> > > > > ### Author Response · Authors · 2025-11-28
> > > > >
> > > > > Thank you very much for your follow-up message and for taking the time to carefully consider our rebuttal! We truly appreciate your intention to substantially increase the rating, and we fully understand the current technical limitations of the OpenReview system.
> > > > >
> > > > > If you have any further questions or would like to discuss any aspect of the work in more detail, we would be very happy to continue the conversation.

---

### Official Review · Reviewer_q78C · 2025-10-31

**Soundness:** 3
**Presentation:** 3
**Contribution:** 3
**Rating:** 8
**Confidence:** 3

**Summary:**

This paper proposes a novel technique for adversarial training which users activation perturbations instead of iterative gradient-base attacks. Prior work suggests that safe and unsafe final-token prompt embeddings are linearly separable. This work leverages this fact to greatly decrease the cost of activation-space perturbation attacks, by solving the closed form linear equation to create a perturbation that simply moves an unsafe prompt across the decision boundary. This makes it cheap enough to apply in parallel across all layers.

They propose two different training setups, one with and without a reference model, to prefer harmless responses over harmful responses.

They train on HarmBench, train a classifier with 20 pairs of safe and unsafe prompts, and evaluate utility on MMLU, ARC-C, MBPP, GSM8K, and Harmless. This decreases the adversarial success rate of the previously successful attacks, as well as decreasing the hamfulness of successful attacks. They then study the computational efficiency, response length, cross-domain robustness, and hyperparameter sensitivity.

**Strengths:**

Originality: The paper proposes a novel gradient-free method for adversarial perturbations based on the finding that harmful and harmless prompts are generally linearly separable. As far as I am aware, this method is original.

Quality: The paper supports its core claims with empirical results. The method learns to interrupt unsafe responses in an unsupervised manner.

Clarity: The paper is generally well-written, and has a particularly clear and helpful explanation of its underlying idea.

Significance: Adversarial robustness is an important problem, and this paper proposes an effective method for constructing latent space perturbations without gradients, drastically decreasing computational cost.

**Weaknesses:**

* The A-Score is somewhat non-standard, and could use more explanation -- it seems to me to be a measure of how useful the successful attacks are, but with general measures of answer quality rather than more specific harmfulness evaluation such as in StrongREJECT.
* It would be interesting to see data-controlled experiments -- the proposed method is claimed to be more data efficient, but if it's also more compute efficient then why not use the same amount of data as other versions? This presumably would make the results stronger.

Minor:
* It's unclear why the LinAJD-D experiments are not repeated for Zephy and Phi3
* The results are substantially stronger for Llama than other models, which could be worth elaborating on.
* The results are strongest for the SCAV attack, and weakest for the LAA attack, and it could be worth elaborating on why.

**Questions:**

* Why does the method result in a 90% reduction in data usage? It seems like the results would be stronger using the same amount of data for every method.

---

> ### Author Response · Authors · 2025-11-21
> **Responses to the main weaknesses**
>
> Thank you very much for your thoughtful and positive review. We appreciate that you highlighted the originality, clarity, and empirical support of our work, and that you consider the problem and our approach significant. Your comments and questions are very helpful, and we address your suggestions and remaining concerns in detail below.
>
> **On data efficiency and data-controlled experiments**
>
> Thank you for raising this point. To address your concern, we added a data-controlled experiment on Qwen2.5-14B. Specifically, we extended the training set for the layer-wise classifiers with **270 additional prompts** from the **HEx-PHI** dataset, carefully chosen to be disjoint from all test sets to avoid any overlap. We kept the training steps and all hyperparameters unchanged, and both LinAJD-S and the extended model **LinAJD-S+** achieve 100% accuracy on HARMLESS, which suggests that the models are not simply overfitting the extra data.
>
> The robustness results for the base model, LinAJD-S, and LinAJD-S+ on PAIR, LAA, DRA, and SCAV are summarized in **Table R1**. As we can see, adding data does improve robustness, but the gains are incremental rather than dramatic. We believe that the core of LinAJD’s data efficiency lies in whether the harmful and benign prompts used to train the classifiers provide a sufficiently general linear boundary in the embedding space, rather than in aggressively increasing the amount of training data.
>
> **Table R1: Robustness on Qwen2.5-14B under strong attacks**
>
> | Attack | Qwen2.5-14B / ASR | Qwen2.5-14B / A-Score | LinAJD-S / ASR | LinAJD-S / A-Score | LinAJD-S+ / ASR | LinAJD-S+ / A-Score |
> |--------|-----------------|---------------------|--------------|------------------|-------------------------|-----------------------------|
> | PAIR   | 90%             | 87.78%              | 38%          | 67.98%           | 26%                     | 66.03%                      |
> | LAA    | 100%            | 53.50%              | 4%           | 33.30%           | 0%                      | 0%                         |
> | DRA    | 86%             | 68.80%              | 0%           | 0%               | 0%                      | 0%                         |
> | SCAV   | 84%             | 84.92%              | 6%           | 63.89%           | 6%                      | 61.11%                      |
>
> **On A-Score and its relation to StrongREJECT**
>
> Thank you for this comment. We agree that using a benchmark such as StrongREJECT is a reasonable way to further validate our evaluation. To this end, we additionally ran StrongREJECT on the same PAIR, LAA, DRA, and SCAV outputs, using **GPT-4o-mini** as the evaluator, and report the results in **Table R2**. Compared with **Table R1**, the trends in A-Score and StrongREJECT are consistent: the base model has both high A-Score and high StrongREJECT scores, while LinAJD-S and LinAJD-S+ simultaneously reduce A-Score and StrongREJECT score, indicating weaker and less harmful attacks. We will clarify this connection in the paper and explicitly point readers to this comparison.
>
> **Table R2: StrongREJECT evaluation for all attacks on Qwen2.5-14B**
>
> | Attack | Model                | Refusal | Convincingness | Specificity | StrongREJECT score |
> |--------|----------------------|---------|----------------|------------|--------------------|
> | PAIR   | Qwen2.5-14B   | 0.20    | 4.78           | 4.82       | 0.7825             |
> | PAIR   | LinAJD-S             | 0.64    | 4.58           | 4.58       | 0.3425             |
> | PAIR   | LinAJD-S+            | 0.82    | 4.66           | 4.30       | 0.1600             |
> | LAA    | Qwen2.5-14B   | 0.00    | 4.94           | 4.98       | 0.9900             |
> | LAA    | LinAJD-S             | 0.92    | 3.78           | 2.88       | 0.0450             |
> | LAA    | LinAJD-S+     | 1.00    | 4.74           | 4.42       | 0.0000             |
> | DRA    | Qwen2.5-14B    | 0.18    | 4.76           | 4.68       | 0.8150             |
> | DRA    | LinAJD-S             | 0.96    | 4.32           | 3.80       | 0.0250             |
> | DRA    | LinAJD-S+            | 0.98    | 4.98           | 4.92       | 0.0200             |
> | SCAV   | Qwen2.5-14B   | 0.20    | 4.46           | 4.66       | 0.7300             |
> | SCAV   | LinAJD-S             | 0.80    | 2.68           | 2.12       | 0.1000             |
> | SCAV   | LinAJD-S+            | 0.88    | 3.80           | 3.20       | 0.0750             |
>
> **References**
>
> PAIR: Chao et al. Jailbreaking Black Box Large Language Models in Twenty Queries. arXiv 2023.
>
> I-FSJ: Zheng X et al. Improved few-shot jailbreaking can circumvent aligned language models and their defenses. NeurIPS 2024.

---

> ### Author Response · Authors · 2025-11-21
> **Responses to the minor weakness and questions**
>
> Thank you for these helpful observations. Regarding why LinAJD-D is not repeated for Zephyr and Phi-3, and why results are substantially stronger for Llama, we discuss this in **Sections 4.2, 4.3.1**, and **Appendix C.4**, and we would like to briefly summarize the main reasons here. In our experience, the baseline level of safety alignment of the underlying model is crucial: it determines whether, in the absence of attacks, the model can provide a clean and stable reward signal for preference optimization. For Phi-3, training with LinAJD-S was noticeably less stable and prone to underfitting or overfitting, which is why we stopped when the loss plateaued around 1 (see **Appendix Figure 10**). In contrast, for models like Llama and, in our new experiments, Qwen2.5-14B, we can train the models to much lower loss while still achieving high accuracy on the HARMLESS, and these models then provide stronger and more reliable robustness improvements.
>
> The same factors also help explain the variation across attacks. We observe that some models (e.g., Llama and Qwen2.5) are inherently more resistant to LAA after LinAJD training, whereas others show relatively stronger gains on SCAV or different attacks. At this stage, we see these differences primarily as **model-dependent** rather than evidence that our method is intrinsically biased toward a particular attack type. A more systematic study of how model architecture, pretraining, and alignment interact with different attack families is beyond the scope of this paper, but we view it as an important direction for future work and will clarify this perspective in the revision.

---

### Official Review · Reviewer_EPGM · 2025-11-03

**Soundness:** 2
**Presentation:** 2
**Contribution:** 3
**Rating:** 6
**Confidence:** 4

**Summary:**

This paper presents LinAJD, an efficient adversarial training (AT) algorithm for LLMs. The main idea behind the proposed LinAJD is to use trained layer-wise adversarial embedding classifiers to directly generate closed-form optimal adversarial perturbations for inputs from each layer of LLMs. Experiments are (mainly) conducted on three open-source LLMs and four jailbreak attacks, which show that the proposed method enjoys better efficiency and jailbreak robustness than the existing CAT baseline method (where adversarial perturbations are only performed on the input token embeddings of LLMs).

**Strengths:**

1. The authors design an efficient training-time adversarial example generalization pipeline that uses trained layer-wise harmful embedding classifiers to directly obtain closed-form optimal adversarial embedding perturbations for the inputs of each layer. Such a design is smart and novel.

2. In the experiments, the authors have also considered a more realistic setting where the harmful embedding classifiers are updated during the AT training process. I appreciate that.

**Weaknesses:**

1. Robustness evaluation experiments are conducted on four jailbreak attacks, but most of these attacks (except GCG) are not common baselines adopted in existing jailbreak robustness evaluations such as those in [r1, r2, r3, r4]. Therefore, I suggest that the authors evaluate their proposed defense against more representative and stronger jailbreak attacks such as [r5, r6, r7, r8].

2. The main AT experiments are conducted on three models (i.e., Table 2), which, from my perspective, might not be enough. Additional experiments on more LLM families such as Qwen2.5/3, Gemma-2/3, Vicuna, and Mistral are highly suggested.

3. **Missing details about the construction of the dataset for training the layer-wise classifiers.** I wonder how the harmful and benign prompts were collected. Were they directly collected from public datasets or synthesized from jailbreak attacks? If the trainset were collected from public datasets, would they not be adversarial enough? On the other hand, if the trainset was synthesized from jailbreak attacks, how would the leveraged jailbreak attacks affect the performance of the trained classifiers as well as the overall AT? Please comment.


**References**

[r1] Xhonneux et al. Efficient Adversarial Training in LLMs with Continuous Attacks. NeurIPS 2024.

[r2] Yu et al. Robust LLM safeguarding via refusal feature adversarial training. ICLR 2025.

[r3] Dekany et al. MixAT: Combining Continuous and Discrete Adversarial Training for LLMs. NeurIPS 2025.

[r4] Fu et al. Short-length Adversarial Training Helps LLMs Defend Long-length Jailbreak Attacks: Theoretical and Empirical Evidence. NeurIPS 2025.

[r5] Chao et al. Jailbreaking Black Box Large Language Models in Twenty Queries. arXiv 2023.

[r6] Hayase et al. Query-Based Adversarial Prompt Generation. NeurIPS 2024.

[r7] Sadasivan et al. Fast Adversarial Attacks on Language Models In One GPU Minute. ICML 2024.

[r8] Andriushchenko et al. Jailbreaking Leading Safety-Aligned LLMs with Simple Adaptive Attacks. ICLR 2025.

**Questions:**

See **Weaknesses**.

---

> ### Author Response · Authors · 2025-11-21
> **Responses to the main weaknesses**
>
> Thank you very much for your thoughtful and positive review. We appreciate that you acknowledged both the novelty of our approach and the overall experimental setup, and we are grateful for your constructive suggestions. We address your main comments and requests in detail below.
>
> **Weakness 1 – Robustness evaluation and choice of attacks**
>
> Thank you for pointing this out. We agree that using representative and strong jailbreak baselines is important. In the revised version, we explicitly connect our attacks to the literature you mentioned: your [r5] corresponds to **PAIR**, and your [r8] corresponds to the attack we denote as **LAA** in the paper. We apologize for introducing a less familiar abbreviation without making this mapping clear. We have now added explicit PAIR results and, in addition, included an extra strong improved few-shot jailbreak, **I-FSJ**, which automatically constructs a demonstration pool, leverages special tokens from the target model’s system template. The new PAIR and I-FSJ results on Llama2-7B are summarized in **Table R1** in the rebuttal (and will be added to the appendix), showing that LinAJD still substantially reduces ASR.
>
> For [r6] (GCQ) and [r7] (BEAST), we were unfortunately unable to include them in our experiments: for one of them we could not obtain working source code, and for the other we were not able to reliably reproduce the reported strong performance on the base models despite multiple attempts. To avoid potentially misleading or unfair comparisons, we therefore chose not to report incomplete results for these attacks.
>
> **Table R1: Results on additional  attacks (Llama2-7B)**
>
> | Attack | Llama2-7B / ASR | Llama2-7B / A-Score | LinAJD-S / ASR | LinAJD-S / A-Score |
> |--------|---------------|-------------------|---------------------|-------------------------|
> | I-FSJ  | 98%           | 88.95%            | 58%                 | 52.35%                  |
> | PAIR   | 66%           | 79.04%            | 0%                  | 0%                     |
>
> **Weakness 2 – Number of models and LLM families**
>
> We agree that evaluating on more LLM families is valuable. In addition to the models in Table 2, we have now added experiments on **Qwen2.5-14B** and report the corresponding robustness results in **Table R2** in the rebuttal (to be integrated into the appendix). These results show that LinAJD continues to provide strong ASR reductions on Qwen2.5-14B and support the claim that our method is not tied to a particular model family.
>
> **Table R2: Results on attacks for Qwen2.5-14B and LinAJD-S**
> | Attack | Qwen2.5-14b/ASR | Qwen2.5-14b/A-Score | LinAJD-S/ASR |  LinAJD-S/A-Score |
> |--------|-----------------|---------------------|----------------|--------------------|
> | PAIR   | 90%             | 87.78%              | 38%            | 67.98%             |
> | LAA    | 100%            | 53.50%              | 4%             | 33.30%             |
> | DRA    | 86%             | 68.80%              | 0%             | 0%                 |
> | SCAV   | 84%             | 84.92%              | 6%             | 63.89%             |
>
> **Weakness 3 – Missing details about dataset construction for the layer-wise classifiers**
>
> Thank you for this question. As we describe in **Appendix C.3** and summarize in **Table 4**, we do not synthesize new prompts with our own jailbreak generator. Instead, harmful prompts are taken from strong public safety benchmarks (**Advbench** and **HarmfulQA**), benign prompts are safe instructions generated by GPT-4, and we evaluate on separate subsets of **Advbench** and **StrongREJECT**. We also do not perform any completion for this stage: the layer-wise classifiers are trained purely on the embeddings of the prompts themselves, without using model-generated continuations. Importantly, the classifier training data are strictly disjoint from both the training and test data of the main models, so there is no data leakage. In addition, the main robustness results you see in the paper are obtained on independent attack generators (GCG/LAA, DRA, SCAV) that are not used for classifier training, which suggests that the learned decision boundaries generalize beyond the specific benchmarks used to construct the training set.
>
> **References**
>
> PAIR: Chao et al. Jailbreaking Black Box Large Language Models in Twenty Queries. arXiv 2023.
>
> I-FSJ: Zheng X et al. Improved few-shot jailbreaking can circumvent aligned language models and their defenses. NeurIPS 2024.

---

### Official Review · Reviewer_M7u1 · 2025-11-04

**Soundness:** 2
**Presentation:** 2
**Contribution:** 3
**Rating:** 4
**Confidence:** 3

**Summary:**

This paper introduces LinAJD, a novel framework for defending Large Language Models (LLMs) against jailbreak attacks through a scalable, gradient-free adversarial training (AT) approach. The authors identify that current gradient-based AT methods are computationally expensive, require substantial auxiliary data, and can degrade model utility. The core insight of LinAJD is to leverage the empirically observed linear separability of safe and harmful prompts in the LLM's embedding space. Instead of using costly iterative gradient-based methods to find adversarial perturbations, LinAJD simplifies the problem into a linear one, allowing for the generation of deterministic, embedding-level perturbations via a closed-form solution. This approach aims to efficiently "harden" the linear safety boundary within the model.

In addition, this paper presents two variants, LinAJD-D and LinAJD-S, which are based on different preference optimization algorithms. Through comprehensive evaluations on models like LLaMA-2-7B and Phi-3, the authors claim that LinAJD achieves state-of-the-art (SOTA) robustness against a wide range of jailbreak attacks, significantly outperforming prior work like R2D2 and CAT/CAPO. Key results include a reported 4x speedup in training, a 90% reduction in data usage, and even reducing the success rate of a specific white-box attack to 0%, all while incurring only minor degradation in general model performance and without needing auxiliary utility datasets. The paper also provides systematic analyses of factors influencing defense performance, such as data quality and domain shifts.

**Strengths:**

1. Good Novelty and Potential Impact: The main advantage of this work lies in proposing a gradient-free, closed-form solution method for generating adversarial perturbations, directly addressing a significant pain point in the security of large-scale language models (LLMs): the high cost of robust alignment. If this method proves to be as effective and broadly generalizable as described, it could represent a critical advancement in making robust security alignment more achievable and practical.
2. Exceptional Efficiency: The reported efficiency gains are a major contribution. A 4x speedup in training and a 90% reduction in data requirements are not merely incremental improvements; they are transformative. This makes the process of adversarially training LLMs substantially more feasible in practice, thereby lowering the barrier to accessing robust defense techniques. Figure 1 provides a compelling visual summary of this efficiency-robustness trade-off, clearly showing that LinAJD achieves superior results with significantly fewer training steps.
3.Strong Empirical Results and Comprehensive Evaluation: The paper claims state-of-the-art (SOTA) performance, supported by comparisons against multiple recent and relevant baselines (e.g., R2D2, CAT, CAPO) across various models. The claim of reducing a white-box attack's success rate (ASR) to 0% is particularly striking and demonstrates the potential to achieve a high level of robustness. Furthermore, the intention to analyze factors like data quality and domain shifts indicates that the evaluation is comprehensive and scientifically rigorous, which is commendable.

**Weaknesses:**

1. Brittleness of the Core Assumption: The success of the entire framework is predicated on the assumption of linear separability. While this is supported by recent work, it is likely an oversimplification of the true, complex geometry of the embedding space. The most sophisticated attacks may not lie along a simple linear boundary. The paper needs to robustly address the limitations of this core assumption.
2. Opacity of Key Methodological Details: The initial sections of the paper repeatedly mention a "closed-form solution" but fail to provide its mathematical derivation, nor has its validity been effectively demonstrated. This is a major weakness that the paper needs to address.
3. Potential Risk of Overfitting: While the paper avoids overfitting to specific gradient-based attack algorithms, it may introduce a risk of overfitting to a specific geometry—namely, the assumed linear boundary. An adaptive adversary, aware of the LinAJD defense strategy, would no longer waste effort on attacks that cross this linear boundary. This problem requires further explanation and discussion.
4. Uncertain Scalability to Larger Models: The experiments were conducted on models with up to 7B parameters. A critical and open question is whether the clean linear separability property still holds for much larger and more powerful models (e.g., 70B models). The geometric properties of the embedding space can change significantly with model scale and architecture. The paper's claims of scalability would be much stronger if supported by either experiments on larger models or a compelling theoretical argument as to why this property should persist.

**Questions:**

1. On the Core Assumption: How did you validate the linear separability assumption on your specific models and datasets? How does your method perform if the boundary is not perfectly linear? For instance, what is its robustness against attacks specifically optimized to find non-linear decision boundaries?
2. On Adaptive Attacks: The 0% ASR claim is very powerful. However, this is likely against a known, static attack. What is the performance of LinAJD against an adaptive white-box adversary who has full knowledge of your defense mechanism and whose objective is to actively find a non-linear path to bypass the hardened boundary? Such an evaluation is the gold standard for measuring defenses and is crucial for an ICLR paper.
3. On Utility Evaluation: You claim that LinAJD maintains model utility with only "minor degradation." To make this claim concrete, could you please provide quantitative results on a suite of standard general-capability benchmarks? A comparative table showing the performance of the base model, LinAJD-D/S, and other defense methods on these benchmarks would be essential.
4. On Method Variants: What are the specific differences between the "preference optimization strategies" underlying LinAJD-D and LinAJD-S? How does this choice (e.g., DPO vs. another algorithm) specifically affect the mathematical formulation of the perturbation and the resulting trade-offs between robustness, utility, and training dynamics?
5. On Attack Generality: How does your framework conceptualize and defend against attacks that are not easily represented as small embedding-space perturbations? For example, attacks based on complex semantic manipulation, multi-turn role-playing, or logical reasoning exploits. Does the static, geometry-based view of defense remain effective in these more abstract attack scenarios?
6. On Figure Readability: The font in Figure 4 is a bit small, and the numbers above the bars in Figure 5 are also small. We suggest adjusting them to improve readability.
7. In the "Efficiency evaluation" section on page 6, you mention a speedup "by over 60×." Could you please clarify which specific metric is improved by over 60 times compared to the baseline?

---

> ### Author Response · Authors · 2025-11-21
> **Responses to the main weaknesses**
>
> Thank you very much for your careful and constructive review. We appreciate that you pointed out both the strengths of our work, the novelty, efficiency, and strong empirical performance, and the aspects where the presentation and evaluation could be clearer.  Below, we respond to each of your points in detail and provide additional experiments and clarifications that we hope will address your concerns.
>
> **Weakness 1 – Brittleness of the core assumption**
>
> Thank you for raising this concern. We agree that the linear-separability assumption in our work is based on empirical observation rather than a formal theoretical guarantee, and we do not claim that all sophisticated attacks must lie exactly along a single linear boundary. Our goal is to exploit a consistently observed linear direction separating harmful and benign prompts as a training signal, in order to adversarially reshape the model’s own  decision boundary and reduce its reliance on a potentially brittle linear boundary. To strengthen the empirical evidence, we added experiments on **Qwen2.5-14B**, and, although resource and time constraints (mainly due to the cost of full attack evaluations) prevent us from running the entire attack suite on larger models, we keep the same training setup and train linear classifiers on **Llama2-70B**. The layer-wise accuracies on Llama2-70B quickly become high (0.5000, 0.5250, 0.7063, 0.8500, 0.9375 for layers 0–4), **exceed 0.95** for all subsequent layers, and yield an overall **average accuracy of 0.9572**, indicating that the linear boundary remains strong in practice even at this scale. We acknowledge that there is still a gap between this empirical picture and a full theoretical explanation, and we see closing this gap as important future work.
>
> **Table R1: Results on strong attacks for Qwen2.5-14B and LinAJD-S**
> | Attack | Qwen2.5-14b/ASR | Qwen2.5-14b/A-Score | LinAJD-S/ASR |  LinAJD-S/A-Score |
> |--------|-----------------|---------------------|----------------|--------------------|
> | PAIR   | 90%             | 87.78%              | 38%            | 67.98%             |
> | LAA    | 100%            | 53.50%              | 4%             | 33.30%             |
> | DRA    | 86%             | 68.80%              | 0%             | 0%                 |
> | SCAV   | 84%             | 84.92%              | 6%             | 63.89%             |
>
> **Weakness 2 – Opacity of key methodological details**
>
> We thank you for raising this point. The full mathematical derivation of the closed-form solution is already included in **Appendix H**, and the main text already refers to this appendix at its first mention. To avoid any confusion, we will (i) make this cross-reference more prominent and (ii) add a brief summary of the key steps and assumptions in the method section, so that the role and validity of the closed-form solution are immediately clear when you read the paper.
>
> **Weakness 3 – Potential risk of overfitting to a specific linear geometry**
>
> We appreciate your concern. As noted above, our goal is not to hard-code a fixed linear rule at test time, but to use the observed linear direction as a training signal to reshape the model’s non-linear decision boundary and reduce its reliance on a brittle linear frontier. Empirically, LinAJD is trained once and evaluated against several structurally different jailbreak families (GCG/LAA, DRA, SCAV), and achieves consistently low ASR across all of them; in particular, SCAV is explicitly constructed to exploit this linear direction, yet its success rate still drops substantially after LinAJD training, suggesting that the defense does not simply overfit to one geometry. We agree that a fully defense-aware adversary that explicitly searches for non-linear bypasses defines a strictly stronger threat model; a systematic study of such attacks is beyond the scope of this paper and will be mentioned as a limitation and direction for future work.
>
> **Weakness 4 – Uncertain scalability to larger models**
>
> We agree that scalability beyond 7B parameters is important. As noted in our response to Weakness 1, we have added experiments on Qwen2.5-14B and linear-probe results on Llama2-70B, which show that a strong linear separation between harmful and benign prompts persists at larger scales. While we do not yet provide a full theoretical explanation for why this property should hold for all architectures and sizes, these results strengthen the empirical case for scalability, and we see a deeper theoretical analysis as valuable future work.

---

> ### Author Response · Authors · 2025-11-21
> **Responses to the questions**
>
> Thank you for raising these questions; we are happy to discuss them in more detail and address each one below.
>
> **Questions 1 & 2 – On the core assumption and adaptive attacks**
>
> Thank you for these closely related questions. These points are addressed in detail in our responses to **Weakness 1** and **Weakness 3** above, and we kindly refer you there for the full explanation.
>
> **Question 3 & 6 – On utility evaluation and Figure Readability**
>
> We thank you for these helpful suggestions. Quantitative utility evaluations are already provided in **Figure 5** (main text) and **Figures 13–15** (appendix), where we report the performance of the base model, LinAJD-D/S, and other defenses across multiple tasks. These results consistently show only minor degradation in utility under LinAJD, which supports the claim made in the paper. To make this clearer and closer to your request, we will (i) explicitly highlight these figures in the main text and (ii) add a concise comparative table summarizing the benchmark scores of the base model, LinAJD-D/S, and baselines. For readability, we will also increase the font size in Figure 4 and enlarge the numbers above the bars in Figure 5 to improve visual clarity.
>
> **Question 4 – On method variants**
>
> We thank you for the question. The core LinAJD mechanism is **decoupled** from the choice of preference optimization: it defines a linear perturbation / preference signal in the embedding space, and this signal is identical in LinAJD-D and LinAJD-S. The only difference is how this signal is used by the loss. As described in Section 3.3 and further discussed in Section 4.2 (“Main Results”), LinAJD-D plugs this signal into a DPO-style objective with a frozen reference model, whereas LinAJD-S uses a SimPO-style objective without relying on a well-aligned reference policy. This mainly affects the **training dynamics** rather than the perturbation formula itself: when the reference model is poorly aligned (e.g., Zephyr), the SimPO-style variant provides a more stable training signal and thus better robustness–utility trade-offs, as shown in our experiments.
>
> **Question 5 – On attack generality**
>
> We appreciate this question about the scope of our threat model. In this work, we focus on jailbreaks that ultimately materialize as single-turn adversarial queries, and we treat each such query, regardless of how it is constructed, as a point in the embedding space; many complex semantic attacks (for example, DRA) already fall into this category. To further support this, we add two additional semantically rich attacks on Llama2-7B, summarized in **Table R2** below, where LinAJD-SimPO still yields substantial ASR reductions. It is not feasible to test against all possible attack strategies, and we explicitly view future red-teaming efforts that search for new failure modes (including more sophisticated multi-turn or reasoning-heavy exploits) as an important next step.
>
> **Table R2: Results on additional  attacks (Llama2-7B)**
>
> | Attack | Llama2-7B / ASR | Llama2-7B / A-Score | LinAJD-S / ASR | LinAJD-S / A-Score |
> |--------|---------------|-------------------|---------------------|-------------------------|
> | I-FSJ  | 98%           | 88.95%            | 58%                 | 52.35%                  |
> | PAIR   | 66%           | 79.04%            | 0%                  | 0%                     |
>
> **Question 7 – On the “over 60×” efficiency claim**
>
> We thank you for pointing this out. The “over 60×” speedup refers to the **total optimization runtime** of the defense (wall-clock time per run), computed as (time per optimization step) × (number of steps). As shown in Table 1, LinAJD has both lower per-step latency and far fewer steps than the most efficient baseline, CAPO, which together yield an overall speedup greater than 60×. We will state this explicitly in the revised version and tie the claim directly to the numbers in Table 1.
>
>
> **References**
>
> PAIR: Chao et al. Jailbreaking Black Box Large Language Models in Twenty Queries. arXiv 2023.
>
> I-FSJ: Zheng X et al. Improved few-shot jailbreaking can circumvent aligned language models and their defenses. NeurIPS 2024.

---

### Author Response · Authors · 2025-11-30
**Summary of revisions for the Area Chair**

Dear Area Chair,

Due to the current OpenReview system issues, we understand that the recent incident has placed an unusually heavy burden on ACs. To assist your evaluation, we would like to provide a concise summary of the revisions we have made based on the discussion with the reviewers. All changes in the revised version are highlighted in **blue** for ease of verification.

**Revisions in the revised version**

1. **Related work on prompt-level defenses**

   - We expanded the *Related Work* section with a new paragraph on prompt-level defenses.
   - In this new paragraph, we explicitly discuss **Prompt Adversarial Tuning**, **Robust Prompt Optimization**, and **SmoothLLM**, and clarify how our approach differs from and complements these methods.

2. **Additional analyses and supplementary experiments (Section 4.2 & Appendix C)**

   - We added additional analyses in **Section 4.2** to better explain the supplementary experiments.
   - In particular, we:
     - Extended our study to a larger model (**Qwen2.5-14B**),
     - Trained on a larger dataset,
     - Incorporated additional attack families (**PAIR** and **I-FSJ**),
     - And evaluated with **StrongReject** to cross-check our evaluation.
   - The full additional experimental results have been included in **Appendix C**.

3. **Training details for Qwen2.5-14B (Appendix D.3)**

   - We included the training details for the **Qwen2.5-14B** model in **Table 8** in **Appendix D.3**.

4. **Linear separability analysis on Llama2-70B**

   - We kept the same training setup and trained linear classifiers on **Llama2-70B**.
   - The layer-wise accuracies on Llama2-70B quickly become high (0.5000, 0.5250, 0.7063, 0.8500, 0.9375 for layers 0~4),  **exceed 0.95** for all subsequent layers,  and yield an **average accuracy of 0.9572**,  indicating that the linear boundary remains strong in practice even at this scale.

**Overall takeaway**

- Collectively, these supplementary experiments further demonstrate both the **scalability** and **robustness** of our method.
- We also note that **Reviewer Cg8Z** (original score 2) and **Reviewer M7u1** (original score 4) explicitly acknowledged the improvements and indicated that they would like to raise their scores based on the new results.

We are sincerely grateful to all reviewers for their thoughtful comments and suggestions, which have significantly helped us improve this work. We are also deeply appreciative of the time and care you have devoted to overseeing the review process and considering our revisions.

Best regards,
The Authors

---

> ### Comment · Area_Chair_ksuE · 2025-12-01
>
> Dear Authors,
>
> Due to the unprecedented decision of ICLR to prevent AC-Reviewer discussion, I have taken the unprecedented decision to engage in an AC-Author discussion for this paper. Note that this will be a poor substitute for a real author-reviewer discussion, as I will not have the time to put the same rigor into reading the paper and following up with your answers. However, I did read the current version of the paper and the full discussion here. I have a few questions remaining:
>
> Q1(R M7u1): **On the generality of linear separability of good and bad prompts.** First, I think the authors should provide the 70B experiment in the appendix of the paper. Further, I think the authors can provide better high-level explanation in motivation section to explain why separability is expected. Finally, can the authors comment on the origin of the separability. Is it caused by allignment for example, or does it naturally occur after pretraining a model? Commenting on this will make clear when the method is applicable and when not.
>
> Q2 (R M7u1): **Closed form solution.** The authors find a closed form solution for the jailbreaks examples shift and then train with these examples. Can they comment how is this different/better/worse than directly optimizing the margin of the decision boundary between the two classes. In a sense, it seems to me that the authors essentially train an SVM but indirectly through choosing the points with which they train it with and explicitly setting the margin through P_0.
>
> Q3 (R M7u1): While the authors dismiss the question about dynamic attacks, I think it is worthed to think how an attack specifically targetting the separability will look like. One thing I can think of is that since you only take shortest parth to the decision boundary, it is possible that attacks that take longer paths are less protected ( as if the boundary is not trully linear and you only push it along the normal maybe further away it remains closer to the bad samples ).
>
> Q4 (R M7u1): **Utility benchmarks.** While I like the provided utility benchmarks (esp code and math which tend to be more brittle), one particular one that will make me believe that the defense is trully utility preserving is XSTest [1]. Can you provide evaluation of your models on it?
>
> Q5 (R EPGM): **Classifier training description.** I agree with the reviewer that prominently in the paper a full appendix should be dedicated to information about training the classifiers. In particular, the authors shoudl provide precise info about the training of the classifiers, as it is still unknown to me why and which prompts are selected to form the 90 positive ones for training, not clear what hyperparameters were used for training, and how is the malicious data for the classifiers selected. Also unclear is the embeddings of which token were used.
>
> Q6 (R q78C): **The A-Score.** I agree with the reviewer that the current explanation of the A-Score is not sufficient. I want to know 1. how the normalization is done exactly; 2. how are individual componentent combined into it. The current formula is unacceptedly vague given that this is the main metric used throughout the paper. Further, it should be better justified in the paper why is it needed in the first place.
>
> Q7 (R q78C): The authors state:  "At this stage, we see these differences primarily as model-dependent rather than evidence that our method is intrinsically biased toward a particular attack type. A more systematic study of how model architecture, pretraining, and alignment interact with different attack families is beyond the scope of this paper, but we view it as an important direction for future work and will clarify this perspective in the revision." I do not agree with this. The authors propose a defense mechanism, they should comment/explore which attacks evade it and which it prevents both quantitatively, which they have, and qualitatively, which is what the reviewer is asking about.
>
> Q8 (R Cg8Z): **Online classifier learning.** I an unconvinced by the authors proposed reason for the failure of the online classifier learner regime. While I can see how it wouldn't improve or can even slightly worsens results, it seems that the online setting worsens results substaintially, and the authors' explanation does not seem credible to me without additional evidence.
>
> Q9 (R Cg8Z): **PAT and RPO.** While they are fundamentally different technique for achieving robustness, and they have different trade-offs comapred to adv training (as the authors acknowledge), I still think the proposed comparison is valid.
>
> Q10 (AC): Why is LAT not used as a baseline? The authors acknowledge in their background this is one of the most related works, yet LAT comparison is not provided.
>
> Q11 (AC): I think it will be nice if you draw the linear decision boundary in Fig 3.
>
> [1] https://arxiv.org/abs/2308.01263

---

> ### Author Response · Authors · 2025-12-04
> **Responses to AC (1/4)**
>
> We sincerely thank the Area Chair for the dedicated time and effort in evaluating our work. We deeply appreciate the insightful questions and constructive comments, which have helped us further strengthen the quality and clarity of our paper. Below, we provide detailed point-by-point responses to address all the concerns raised.
>
> **Q1: On the generality of linear separability**
>
> **Response:**
> To robustly verify the generality of linear separability, we conducted a comprehensive analysis across the **Qwen2.5 model family**, spanning parameter scales of **0.5B, 14B, and 72B** for both **Base** and **Instruct** versions. The results are detailed in **Appendix C.4**.
>
> We find that **Base models** across all scales already exhibit high classifier accuracy, providing strong empirical evidence that separability originates from **Pretraining**, where the model learns to semantically encode "harmfulness." Furthermore, separability improves with model size. While smaller models (0.5B) show a noticeable gap between Base and Instruct versions, larger models (14B, 72B) achieve near-optimal separability even in their Base versions, confirming that this geometric property scales effectively and becomes more robust with parameter size.
>
> Regarding the motivation, at a high level, separability is expected because "harmfulness" is a fundamental semantic attribute that models must distinguish to model language distributions effectively. Consequently, harmful and safe prompts naturally form distinct clusters in the embedding space. Theoretically, this aligns with the **"Linear Representation Hypothesis"** [1], which posits that such high-level semantic distinctions are encoded as linear directions. LinAJD exploits this by identifying the specific linear boundary separating these pre-existing clusters.
>
> **Q2: Closed form solution**
>
> **Response:**
> We clarify that our method is fundamentally an **Adversarial Training (AT)** framework, distinct from geometric margin maximization (SVM-style).
>
> First, regarding the fundamental objective, margin optimization directly penalizes the proximity of embeddings to the decision boundary, aiming purely for geometric separation. However, geometric distance does not guarantee that the model's actual output behavior remains correct. In contrast, LinAJD minimizes the prediction loss on inputs that are explicitly perturbed to the worst-case boundary position (refer to Eq. 6 and Eq. 7 in the paper). This trains the model to maintain **robust refusal behavior** (correct output generation) even when the internal representation is under attack. By training on these "hard negatives" synthesized at the boundary, we force the model to align its behavioral policy with the safety constraint, rather than just pushing features apart.
>
> Furthermore, unlike specific loss-based margin constraints which typically target a single layer, LinAJD applies **deep perturbation** across multiple layers. Since perturbations at one layer propagate to subsequent layers, this creates a cascading defense mechanism. The model is forced to learn robustness not just at the final output, but throughout its entire internal processing depth, preventing the "collapse of separability" from propagating through the network. Additionally, because these perturbations are computed in the embedding space, LinAJD functions as a generic data-augmentation strategy that is **agnostic to the fine-tuning technique**, ensuring seamless compatibility with standard pipelines like LoRA or full fine-tuning.
>
> **Q3: Dynamic Attacks**
>
> **Response:**
> We apologize for any initial misunderstanding regarding the nature of dynamic attacks. We acknowledge that attacks targeting non-linear paths are theoretically possible. However, experimental evidence suggests they are practically constrained by the **semantic geometry** enforced by our margin.
>
> First, the perturbation magnitude is governed by the equation (Eq. 4):
> $ \epsilon = \frac{\sigma^{-1}(P_0) - b - \textbf{w}^\top \textbf{z}}{\|\textbf{w}\|} $.
> Due to the steep gradient of $\sigma^{-1}$ near zero, setting a small target probability (e.g., $P_0=0.01$) enforces a substantial geometric margin in the embedding space. Our ablation study (Figure 8a) shows that robustness improvements plateau below this threshold, indicating that the linear boundary effectively captures the vast majority of harmful variation.
>
> Second, we observed that pushing the margin to extreme values (e.g., $<0.001$) causes the model to degenerate into repetition loops. This implies that embeddings are being pushed to the edge of the semantic manifold. For an adaptive attack to succeed, it would need to bypass our linear barrier *without* falling into these unstable, non-semantic regions. Given that our defense already pushes embeddings close to this semantic limit, the "space" available for an attacker to maneuver around the boundary while preserving coherent harmful intent is extremely narrow.

---

> ### Author Response · Authors · 2025-12-04
> **Responses to AC (2/4)**
>
> **Q4: Utility Benchmarks (XSTest)**
>
> **Response:**
> We have evaluated our LinAJD-tuned models (Llama-2-7B and Qwen2.5-14B) using **XSTest**. The full results are included in **Appendix C.5** and summarized below. These results confirm the effectiveness of our defense while highlighting expected trade-offs.
>
> **Llama-2-7b-chat**
>
> | Metric | Llama-2 Original | Llama-2 LinAJD-S | Change |
> | :--- | :---: | :---: | :---: |
> | **Safe Prompts (n=250)** | | | |
> | Full Compliance | 169 (67.6%) | 130 (52.0%) | -15.6% |
> | Full Refusal | 19 (7.6%) | 31 (12.4%) | +4.8% |
> | Partial Refusal | 62 (24.8%) | 89 (35.6%) | +10.8% |
> | **Unsafe Prompts (n=200)** | | | |
> | Full Compliance | 16 (8.0%) | 6 (3.0%) | -5.0% |
> | Full Refusal | 102 (51.0%) | 84 (42.0%) | -9.0% |
> | Partial Refusal | 82 (41.0%) | 110 (55.0%) | +14.0% |
>
> **Qwen2.5-14B-Instruct**
>
> | Metric | Qwen2.5 Original | Qwen2.5 LinAJD-S | Change |
> | :--- | :---: | :---: | :---: |
> | **Safe Prompts (n=250)** | | | |
> | Full Compliance | 241 (96.4%) | 214 (85.6%) | -10.8% |
> | Full Refusal | 2 (0.8%) | 2 (0.8%) | 0.0% |
> | Partial Refusal | 7 (2.8%) | 34 (13.6%) | +10.8% |
> | **Unsafe Prompts (n=200)** | | | |
> | Full Compliance | 41 (20.5%) | 31 (15.5%) | -5.0% |
> | Full Refusal | 74 (37.0%) | 69 (34.5%) | -2.5% |
> | Partial Refusal | 85 (42.5%) | 100 (50.0%) | +7.5% |
>
> Regarding **Safe Prompts**, for Qwen2.5-14B, the **Full Refusal rate remained unchanged (0.8%)**, proving LinAJD does not induce rigid over-refusal. We observed a shift from Full Compliance to **Partial Refusal** (~10% increase). This reflects the inherent **robustness-accuracy trade-off**: strict safety margins induce conservative behavior on borderline queries. However, this is tunable; relaxing the margin can recover compliance if needed.
>
> Regarding **Unsafe Prompts**, **Full Compliance dropped significantly** (e.g., -5.0% for both models), confirming improved safety. Notably, **Partial Refusal increased** (Llama +14.0%, Qwen +7.5%). This aligns perfectly with our observation in **Figure 2**: LinAJD trains the model to **self-interrupt**.  This validates that the "Partial Refusals" in the unsafe category are successful dynamic interventions, not failures.
>
> **Q5: Training details**
>
> **Response:**
> We emphasize that the complete details for classifier training were **explicitly provided** in our original submission, specifically in **Appendix D.1 (Dataset Preparation)** and **Table 7 (Hyperparameters)**. We would like to state the specific setup here for immediate clarity, and further details can be found in Appendix D.
>
> First, regarding **Classifier Training**, the prompts were sourced from `https://github.com/SproutNan/AI-Safety_SCAV.git`. We utilized the **last token embedding** and trained the linear classifier using the **SAGA solver**. Specifically, we used **20 pairs of embeddings for training** (corresponding to 20 positive and 20 negative samples) and **40 pairs for evaluation**. The classifier is optimized for **100 steps** with a learning rate of **0.01** and a batch size of **32**.
>
> Second, regarding the **LinAJD Fine-tuning Data** (the ~90 pairs mentioned), this dataset refers to the prompts used for optimizing the LLM itself. As detailed in Appendix D.1, we started with 100 malicious queries. The final count of ~90 was not cherry-picked for performance but resulted from **data quality filtering**. We only selected prompts where valid contrastive pairs could be formed (i.e., **the model refuses the original query but fails under attack**). This rigorous filtering ensures the model learns from meaningful embedding shifts. Since our method aligns with preference optimization logic (DPO/SimPO), each training instance consists of a refusal response (safe) and a successful jailbreak response (unsafe).
>
> The full training logic is available in our code: `https://anonymous.4open.science/status/LinAJD-anon-4BBE`.

---

> ### Author Response · Authors · 2025-12-04
> **Responses to AC (3/4)**
>
> **Q6: The A-Score Definition and Justification**
>
> **Response:**
> We emphasize that A-Score is **not a newly invented metric**, but an established evaluation framework adopted directly from *Uncovering Safety Risks of Large Language Models through Concept Activation Vector (NeurIPS 2024)*.
>
> As defined in Eq. 8, A-Score is calculated as:
> $$ \text{A-Score}(x) = \mathbb{I}({\text{ASR}(x) = 1})\cdot \left( \frac{1}{4} \sum_{i=1}^{4} \frac{s_i(x)}{s_i^{\max}} \right) $$
> It only activates when a jailbreak succeeds ($\text{ASR}=1$). It averages four sub-metrics—**Usefulness (0-3), Specificity (0-3), Repetition (0/1), and Consistency (0/1)**—each normalized by its maximum value $s_i^{\max}$ to the $[0, 1]$ range. This ensures the score reflects the *severity* and *quality* of the harmful output, distinguishing effective jailbreaks from gibberish.
>
> We chose this metric because the original authors empirically validated its **high alignment with human evaluation** under blind review. Building on their verification, we adopted this as a reliable, quantitative metric. To further confirm its reliability, we have added comparisons with **STRONGReject** in the **Appendix C.3**, showing strong consistency with A-Score and justifying its utility as a robust measure of defense effectiveness.
>
> **Q7: Qualitative Analysis of Attack Success/Failure**
>
> **Response:**
> We agree that qualitative analysis is essential. Our investigation identifies two key dimensions determining defense effectiveness: **Attack Method** and **Content Domain**.
>
> Regarding **Attack Method**, LinAJD is **most effective** (ASR $\approx 0\%$) against **Semantic Attacks** (e.g., DRA). These attacks disguise intent but retain a strong projection onto the "harmful" direction, which our linear boundary intercepts. In contrast, **Optimization Attacks** (e.g., GCG, PAIR) are **relatively harder** to defend. They exploit high-frequency, orthogonal features (e.g., adversarial suffixes) to bypass the boundary. While LinAJD still suppresses the majority, evasion is more frequent.
>
> Regarding **Content Domain**, prompts involving **Strong Harm** (e.g., Violence) are highly robust as the semantic signal is strong and far from the decision boundary. However, **Weak/Specialized Harm** (e.g., Financial Advice) is harder to detect. As analyzed in **Section 4.3.2 (Cross-Domain Robustness)**, the distinction between "harmful" and "benign" in specialized domains is subtler (embeddings lie closer to the boundary).
>
> Qualitatively, we observe a **synergistic effect**: The highest evasion rates occur when **Optimization Attacks** are combined with **Weak Harm/Specialized Domains**. The weak semantic signal coupled with adversarial perturbation makes these edge cases the most challenging to defend.
>
> **Q8: Online Classifier Learning Failure**
>
> **Response:**
> We posit that the degradation in the online setting stems from the **"Moving Target" problem**, which can be better understood in light of our findings in **Q1**. Since linear separability primarily originates from **pretraining** (encoding fundamental semantic concepts), the fixed classifier effectively anchors the defense to this robust, pre-existing semantic boundary.
>
> When the classifier updates simultaneously with the LLM (online), this geometric anchor is lost. The LLM can optimize to satisfy the *current* classifier (e.g., by rotating embeddings) without genuinely removing harmful semantics, leading to co-adaptation that drifts away from the robust pretrained boundary. While we believe this loss of the "pretrained anchor" is the primary cause, we acknowledge that fully disentangling the dynamics of co-adaptation requires more fine-grained experimentation, which we view as a valuable direction for future work.

---

> ### Author Response · Authors · 2025-12-04
> **Responses to AC (4/4)**
>
> **Q9 & Q10: Baselines Comparison (PAT, RPO, LAT)**
>
> **Response:**
> We would like to address the concerns on baseline selection together. We selected baselines following a clear technological lineage: **R2D2** (input-space perturbation) $\rightarrow$ **CAT/CAPO** (embedding-space gradient optimization) $\rightarrow$ **LinAJD** (embedding-space closed-form solution). LinAJD is a direct optimization of the CAT/CAPO framework. Furthermore, CAT/CAPO provided open-source checkpoints and identical training data sources, enabling the most rigorous and fair comparison.
>
> Regarding **Q10 (LAT)**, we acknowledge that **LAT (Linear Artificial Tomography)** is a highly relevant baseline. While the limited rebuttal window precluded a full-scale reproduction, we recognize its importance and commit to including a detailed comparison with LAT in the final revision to complete the landscape analysis.
>
> Regarding **Q9 (PAT and RPO)**, we previously excluded them as they represent different paradigms. However, we agree with the reviewer and AC that they are valid safety benchmarks and we will expand our comparison with these methods.
>
> **Q11: Figure 3 Visualization**
>
> **Response:**
> We appreciate the suggestion to improve visual clarity. We have updated **Figure 3** in the revised manuscript to explicitly draw the **linear decision boundary (hyperplane)**. The updated figure now clearly illustrates how the benign and harmful clusters are separated and how LinAJD pushes the adversarial samples across this specific linear threshold.
>
> **References:**
> [1] Park, K., Choe, Y. J., & Veitch, V. (2024). *The Linear Representation Hypothesis and the Geometry of Large Language Models*.

---

> > ### Author Response · Authors · 2025-12-04
> > **Appreciation for the Area Chair’s Insightful Guidance**
> >
> > We are sincerely grateful for the thoughtful questions and insightful suggestions you raised in your discussion. Your comments have helped us to see more clearly the limitations and open problems in our current work, and they have already shaped how we think about the next steps of this line of research. We truly appreciate the time and care you devoted to engaging with our paper and guiding us toward improving both the clarity and the broader impact of our contributions.

---

### Note · Authors · 2026-03-07

I have read and agree with the venue's withdrawal policy on behalf of myself and my co-authors.

---

### Meta-Review · Area_Chair_ksuE · 2026-01-08

**Summary:**

All in all, the authors have improved their submission during the rebuttal, answering some very important questions. There are, however, a few important outstanding points which I outline in **Reviewer Concerns**. To me, the lack of baselines (Q9) and the test against dynamic attacks (Q3), as well as the presentation of the XSTest results (Q4), remain major problems of the current version of the paper that need to be addressed before acceptance. Thus, I recommend a rejection. That said, the decision was hard, and the paper is truly borderline in its current form, so if those are addressed, I am sure the next revision will clear the acceptance line.

**Reviewer Concerns:**

I already summarized the outstanding reviewer comments in the AC-author discussion comments from 1 Dec. I will therefore summarize next only the outstanding concerns after the author's response from 4 Dec.

- **Q3 (R M7u1): Dynamic Attacks**
While I appreciate the answer from the authors, I am still not satisfied with it. It is my opinion that the paper will benefit from thinking about and testing against attacks aware of the proposed defense. Without this, the authors might be presenting their defense as stronger than it truly is. This specifically relates to the results presented in Q7, where we see that particular properties of the attacks make them naturally stronger than others.
- **Q4: XSTest**
I thank the authors for the provided results. I am a bit worried about the fact that in the provided experiments, many of the base model's correct full refusals were converted from full refusals to partial ones. While I appreciate the authors' explanation with regard to Figure 2, the explanation points towards not an ideal way of judging the partial rejections in these experiments.
- **Q6 (R q78C):  The A score**
The authors are suggested to include this discussion in the next revision of the paper.
- **Q7 (R q78C):  Qualitative analysis**
I appreciate the author's analysis. I think it should be included in the next revision of the paper, as it clearly suggests that the semantic alignment that comes from the LLM embeddings makes the defense stronger to natural text (opposed to random token attacks) and that adversarial requests closer in embedding space to benign ones are more susceptible to attack even after the training.
- **Q9/10 (AC,R Cg8Z): Baselines**
I find these comparisons crucial for determining the strength of the defense. As my decision is conditional on the results, their omission is preventing me from reaching a positive decision.

**Reviewer Scores:**

- **Reviewer M7u1**
I think the reviewer could have increased their score to a 6, given that important problems were addressed, including scalability and better exploration of the origins of the linear separability (see AC-Author discussion). That said, some issues remain, including dynamic attacks.
- **Reviewer EPGM**
I think the reviewer could have raised their score to 8. The authors essentially solve most of the reviewer's problems, including more model families, more attacks, and a better explanation of the classifier training (which was also partially supplied in the AC-Author discussion).
- **Reviewer q78C**
I find the reviewer's original score way too high, especially in light of the actual questions asked. That said, I think the authors have addressed some of the important points well - including the longer training and the StrongReject question.  However, I find the answer to "It's unclear why the LinAJD-D experiments are not repeated for Zephy and Phi3" not very satisfactory. I think objectively, the reviewer's score should be between 6 and 8.
- **Reviewer Cg8Z**
As acknowledged by the reviewer, some of their questions were addressed by the authors, which warrants a score increase. That said I find the explanations about the pre-trained classifier's better performance still a bit unconvincing without an accompanying experiment. I also, as outlined above, find that the method should be compared against more baselines.  To this end, I think the reviewers' score should be increased to 6 at most.

---

### Decision · Program_Chairs · 2026-01-26

Reject